# Bacterial Communities in the Feces of Laboratory Reared *Gampsocleis gratiosa* (Orthoptera: Tettigoniidae) across Different Developmental Stages and Sexes

**DOI:** 10.3390/insects13040361

**Published:** 2022-04-07

**Authors:** Zhijun Zhou, Huimin Huang, Xuting Che

**Affiliations:** 1Key Laboratory of Zoological Systematics and Application of Hebei Province, College of Life Sciences, Hebei University, Baoding 071002, China; 17862061092@163.com (H.H.); chexuting_1994@163.com (X.C.); 2Institute of Life Science and Green Development, Hebei University, Baoding 071002, China

**Keywords:** katydids, *Gampsocleis gratiosa*, bacterial community, developmental stage, 16S rDNA V3-V4 region, Illumina sequencing

## Abstract

**Simple Summary:**

Many insects host a diverse gut microbial community, ranging from pathogenic to obligate mutualistic organisms. Little is known about the bacteria associated with katydids. *Gampsocleis gratiosa* (Orthoptera, Tettigoniidae) is an economically important singing pet in China. In the present study, the bacterial communities of the laboratory-reared *G. gratiosa* feces were characterized using Illumina sequencing of the 16S rDNA V3-V4 region.

**Abstract:**

We used Illumina sequencing of the 16S rDNA V3-V4 region to identify the bacterial community in laboratory-reared *G. gratiosa* feces across different developmental stages (1st–7th instar nymph day 0, and 0-, 7-, 14-, and 21-day adult) and sexes. In total, 14,480,559 high-quality reads were clustered into 2982 species-level operational taxonomic units (OTUs), with an average of 481.197 (±137.366) OTUs per sample. These OTUs were assigned into 25 phyla, 42 classes, 60 orders, 116 families, 241 genera, and some unclassified groups. Only 21 core OTUs were shared by all samples. The most representative phylum was Proteobacteria, followed by Firmicutes, Bacteroidetes, and Acidobacteria. At the genus level, *Kluyvera* (387 OTUs), *Obesumbacterium* (339 OTUs), *Buttiauxella* (296 OTUs), *Lactobacillus* (286 OTUs), and *Hafnia* (152 OTUs) were dominant bacteria. The early-instar nymphs harbored a similar bacterial community with other developmental stages, which contain higher species diversity. Both principal coordinate analysis (PCoA) and non-metric multidimensional scaling analysis (NMDS) failed to provide a clear clustering based on the developmental stages and sexes. Overall, we assume that *G. gratiosa* transmits bacteria vertically by eating contaminated eggshells, and both developmental stages and sexes had no significant effect on the fecal bacterial community.

## 1. Introduction

Many insects harbor a diverse gut microbial community, including protists, fungi, archaea, and bacteria [1]. Numerous previous works have demonstrated that these symbiotic microbiotas potentially provide many beneficial services to the host’s overall ecological fitness: e.g., feeding, digestion, nutrient absorption, immunity, growing development, and even insecticide resistance [2,3,4,5,6,7]. For instance, the gut bacterial communities of camellia weevil *Curculio chinensis* are consistent with a potential microbial contribution to the detoxification of the genus *Camellia* tree defensive chemicals [8]. The gut microbial composition may vary widely from insect to insect because of their different feeding habits. On the whole, the gut microbial diversity of omnivorous insects was significantly higher than that of carnivorous and herbivorous insects [9]. For example, omnivorous cockroaches *Shelfordella lateralis*, *Blattella germanica*, and *Periplaneta americana* harbor relatively rich gut microbial species [10,11,12,13]. The majority of gut microbiota were not essential for host insect survival and obtained via the environment rather than via vertical transmission [10]. Large differences in annual temperature and humidity could alter the gut microbial composition and diversity, and assist host insects’ survival in these local environments [14].

The basic insect life cycle presents potential challenges for transmission of microorganisms between generations. The opportunity of direct transfer for gut symbionts between conspecifics are more limited in solitary insects with non-overlapping generations. Females sometimes display sophisticated mechanisms for inoculating eggs or progeny with microbial symbionts [15]. Adult females potentially transmit bacteria to progeny by defecating in the vicinity of eggs and having their gut bacteria ingested by their progeny [1]. Female red firebugs *Pyrrhocoris apterus* (Hemiptera: Pyrrhocoridae) and many lepidopterans transfer microbes vertically by smearing their microbiota on the eggs [16,17]. The stinkbug *Riptortus clavatus* postnatally acquires a beneficial gut symbiont from the environment every generation [18]. Caterpillar gut microbiomes are dominated by leaf-associated bacteria, further suggesting that resident, host-specific symbionts are sparse or absent [19]. Gut symbiotic communities can be dynamic, changing through time and developmental stage [20]. Several larvae–adult sample pairs were analyzed independently to further assess species-specific and diet-specific effects on conspecific individuals of varying life stages [21,22,23,24]. The shedding of foregut and hindgut exoskeletal lining during insect larval development severely disrupts or eliminates any attached bacterial populations [1]. The midgut bacteria of *Drosophila* have been associated with changes in host development [25,26].

Bacterial species comprise all or most organisms of most insect guts [1]. Bacterial diversity studies using culture-independent methods depended on the 16S rDNA gene. The bacterial 16S rDNA full-length [27] and hypervariable regions V1–V3 [12,28], V3–V4 [14,29,30], and V4 [31,32] have been widely used. High-throughput sequencing technologies can detect significantly higher diversity in microbial populations than traditional culture-based and conventional molecular methods [9,29]. Until recently, the Illumina platform had greater potential because it generated longer sequence reads and was more suited for smaller projects [33].

The diversity and composition of bacterial community vary substantially across different developmental stages of holometabolous insects, such as *Dendroctonus rhizophagus* [34], *Monochamus alternatus* and *Psacothea hilaris* [35], and *Spodoptera littoralis* [6]. Orthopterans are hemimetabolous insects with similar nutritional environments and requirements during their developmental stages. The egg hatches into a nymph, which feeds, ecdysis, and grows larger, then emerges as an adult that looks very similar to the late-instar nymph. Previous studies focused mainly on the orthopteran adult gut bacterial community [36,37,38]. Katydids and grasshoppers shared a characteristic bacterial community dominated by Proteobacteria, Firmicutes, and Actinobacteria [39]. The midgut bacteria of desert locust *Schistocerca gregaria* produce phenolics with antimicrobial properties that protect host insects from entomopathogens [39]. Both trophic behaviors and the evolution of the host may contribute to the shifts in prevalence among the core bacterial groups of six grasshopper species [36]. The gut bacterial communities of katydids having different feeding habits were obviously different, in which omnivorous katydids host the highest gut bacterial diversity [38].

Microbial symbiosis involves acquisition, colonization, and transmission. It is poorly understood whether the orthopteran bacterial community varies among the different developmental stages and between sexes. Omnivorous katydids *Gampsocleis gratiosa* (Orthoptera: Tettigoniidae) are well-known as singing pets in China. Hence, we used Illumina sequencing of 16S rRNA V3–V4 regions to assess the bacterial communities in laboratory-reared *G. gratiosa* feces across different developmental stages and sexes. We hypothesize that bacterial diversity increased as *G. gratiosa* progressed through first instar nymph to adult, and shared several core bacteria between different life stages and sexes. To the best of our knowledge, this is the first attempt to study the bacterial communities across the life history of a laboratory-reared katydid.

## 2. Materials and Methods

### 2.1. Insect Rearing and Fecal Sample Collection

Fifty-five newly hatched *G. gratiosa* nymph (N) were reared until reaching 21 d adult (A) after the final adult ecdysis. It was reared in separate cages under a regime of 12 h lightness at 32 °C, 12 h darkness at 18 °C, and 50 ± 5% relative humidity. Both nymphs and adults were fed with a 1:1:1 mixture of steamed carrot, chicken liver, and cooked soya. There are seven instars, and complete nymphal development in about 60 days. Fecal pellets were collected singly in aseptic microtube at 17:00 daily. Afterward, all fecal samples were stored at −80 °C until required. Avoided interrupting *G. gratiosa* molting when collecting. We recorded the molting date in detail.

The fecal samples of 36 *G. gratiosa* individuals (18 females and 18 males) were analyzed at 11 time points: 1st–7th instar nymph on molting day, and 0-, 7-, 14-, and 21-day adult. The feces of 6 *G. gratiosa* individuals were pooled as one sample and three biological replicates were established for each time point. The sample labels and grouping were shown in Table 1.

### 2.2. DNA Extraction, 16S rDNA V3–V4 Amplification and Illumina Sequencing

Bacterial DNA were extracted from each pool using the E.Z.N.A™ Mag-Bind Soil DNA Kit (Omega Bio-tek, Inc. Norcross, GA, USA). Bacterial 16S rDNA V3–V4 hypervariable region was utilized to assess gut bacterial diversity. Amplification was performed using the 314F (5′- CCT ACG GGN GGC WGC AG -3′) and 805R (5′- GAC TAC HVG GGT ATC TAA TCC -3′) primers [40]. Both primers contained Illumina adapters, and the reverse primer contained a 6 bp barcode sequence unique to each sample. Polymerase chain reaction (PCR) was performed using 2 × Taq Master Mix (Vazyme Biotech Co., Ltd. Nanjing, China) with the following conditions: 94 °C 3 min; 30 cycles of 94 °C 30 s, 50 °C 30 s, 72 °C 1 min; 72 °C 7 min. The PCR products were quantified with Qubit3.0 (Qubit^®^ ssDNA Assay Kit, Thermo Fisher Scientific Inc., Shanghai, China) and pooled together in equimolar concentrations for sequencing. High throughput DNA sequencing was conducted using a paired-end, 2 × 300 bp cycle run on an Illumina Miseq^TM^ platform at Shanghai Sangon Biotech Co., Ltd. (Shanghai, China). No technical replicates were performed on samples.

### 2.3. Bioinformatics and Statistical Analysis

Primers were trimmed from raw reads using Cutadapt 1.2.1 [41] and assigned to their respective samples according to the unique barcodes. The sequences were filtered by quality using Prinseq 0.20.4 [42], a Phred quality cutoff value of 20 (Q20) and a minimum cut length of 200 nucleotides were used for both strands. After the removal of barcodes and primers, forward and reverse reads were joined using PEAR v.0.9.10 [43].

Non-amplified region sequences were removed using Usearch 9.2 [44]. Chimeras were detected and removed using Uchime 4.2.40 [45]. Cleaned sequences were pooled and clustered into operational taxonomic units (OTUs) at a 97% similarity level using Usearch 9.2 [44]. OTUs with less than 0.005% of the total number of reads were further interpreted as likely contaminants and filtered from the OTU table [46]. Rarefaction curves of the observed OTUs were constructed in Usearch v10.0.240.

The most abundant sequence of each OTU was chosen as a representative sequence, and taxonomic assignment was made with the Bayesian RDP classifier 2.12 against the Ribosomal Project Database (Confidence level = 0.8) [47] and Silva databases (Similarly > 90% and Coverage > 90%) [48]. We removed the sequences from OTU tables, which were annotated as chloroplasts or mitochondria, and not assigned to any kingdom, to ensure only bacterial 16S rDNA sequences were included in downstream analyses. Relative abundance of each OTU was counted from phylum to genus level [49]. The core bacteria of *G. gratiosa* was defined as those OTUs present in all 66 fecal samples.

All statistical tests were performed in R v4.1.2 (http://cran.r-project.org accessed on 1 January 2022), mainly using vegan v2.5-7 package. The OTU table was subsampled to avoid sequencing depth effect according to the sample with the lowest number of reads (n = 168,005). Alpha and beta diversity analyses were calculated using the vegan package and plotted using the ggplot2 package. The differences among developmental stages and between sexes were analyzed by alpha diversity indices (ACE, Chao1, Shannon, and Simpson) using Kruskal-Wallis (KW) rank test [50] and paired *t*-test, respectively. Differences between compared groups were considered significant when *p* < 0.05. Non-metric multidimensional scaling (NMDS) was used to visualize the sample groupings based on the Bray-Curtis dissimilarity indices [51]. Analysis of similarity (ANOSIM) was performed to determine the statistical significance among groups with 9999 permutations. Principal coordinates analysis (PCoAs) was used to visualize the sample groupings based on the Bray-Curtis dissimilarity indices [51], and unweighted and weighted UniFrac distances [52]. Beta diversity was also performed through heatmap analysis for Top20 genus using a pheatmap (v1.0.12) package. Log-linear discriminant analysis effect size algorithm (LEfSe) [53] analysis was performed to identify potential taxonomic groups (LDA scores > 2.0, *p* < 0.05) that can serve as biomarkers for different groups.

## 3. Results

### 3.1. Sequencing Statistics

Like all orthopteran insects, *G. gratiosa* undergoes incomplete metamorphosis; nymph and adults had hardly any differentiations in form or function (Figure 1A). Illumina sequencing of 16S rDNA V3-V4 region for 66 fecal samples across the *G. gratiosa* life history yielded 14,907,483 raw reads. Following sequence trimming, quality filtering, and removal of contaminants (non-16S rDNA, organelle 16S rDNA and chimeras), remaining 14,480,559 high-quality clean reads were retained. A summary of the number of pre-filtering, post-filtering reads for each fecal sample is given in Appendix A.

### 3.2. Bacterial Community Structures and OTUs

These clean reads fell to 80,492 OTUs at the 97% similarity interval. After removal of OTUs with less than 0.005% of the total number of reads, 2982 OTUs were counted across all samples. In general, these ultra-low abundance OTUs were interpreted as likely contaminants and discarded before the downstream analyses [46,54]. The number of observed OTUs varied from 204 to 899 per sample (Table 1), with an average of 481.197 (±137.366) OTUs per sample. The majority of OTUs (2507 of 2982 OTUs) were present in two or more samples. The species accumulation curves reached an asymptote after ~30 samples, indicating that the number of sequenced samples enough to observe most bacteria in *G. gratiosa* feces (Figure 1B). Rarefaction curves plotted from observed OTUs have reached plateau with a high Good’s coverage index (>99%) that indicated that the sequence depth were able to represent majority of the OTUs present in all the 66 fecal samples (Figure 1C). A higher OTUs count indicated a highly complex bacterial community in *G. gratiosa* feces.

The 2982 OTUs were assigned into 25 phyla, 42 classes, 60 orders, 116 families, 241 genera, and some unclassified groups. The number of unassigned OTUs across different taxonomic ranks was raised from 27 OTUs (representing 0.01% total reads) at phylum level to 860 OTUs (representing 2.07% total reads) at genus level.

The majority of OTUs fell into four major bacterial phyla: Proteobacteria, Firmicutes, Bacteroidetes, and Acidobacteria (Table 2 and Appendix A), while the remaining 21 phyla represented less than 1% OTUs and total reads. Within the Proteobacteria phylum, Gammaproteobacteria, Alphaproteobacteria, and Betaproteobacteria were the most dominant. Within the Firmicutes phylum, Bacilli, and Clostridia were the most dominant. At the order and family level, the Enterobacteriales order all belonged to the Enterobacteriaceae family within class Gammaproteobacteria, while the Lactobacillales order was almost completely represented by the Lactobacillaceae family within class Bacilli. At the genus level, 72 of 241 (29.88%) genera occurred in all eight groups, and seven genera from Proteobacteria and Firmicutes were dominant (relative abundance >1%) (Table 2).

Nine of 25 phyla Acidobacteria, Actinobacteria, Bacteroidetes, Chloroflexi, Euryarchaeota, Firmicutes, Proteobacteria, Thermotogae, and Verrucomicrobia were detected in all eight groups. The majority of OTUs belonged to the most abundant phylum Proteobacteria (ranging from 71.76% in MA to 83.46% in the MN12 group), followed by Firmicutes (ranging from 10.80% in MN12 to 16.77% in the FA group). Bacteroidetes was the third most abundant phylum in all groups except for the FN67 group. Proteobacteria phylum had the highest relative abundances in all groups, except for the FN35 group (53.99% reads of Firmicutes higher than 45.84% reads of Proteobacteria). The remaining seven phyla had lower relative abundance (<1% reads of any sample). The relative abundance of the *Kluyvera*, *Lactobacillus*, and *Hafnia* genus were the top five genera in all groups. These three genera made up 2% or higher reads of any groups. The relative abundance of *Kluyvera* OTU3 in early-instar nymphs was apparently higher than other developmental stages (Appendix A).

### 3.3. Dominant and Core Bacterial OTUs

Thirty-two dominant OTUs (representing >1% reads of any sample) represented 91.56% of total reads. Among these, 24 OTUs and 8 OTUs belonged to the Gammaproteobacteria class of Proteobacteria and the Bacilli class of Firmicutes, respectively. Within Gammaproteobacteria, 24 OTUs were assigned to the genus *Aeromonas* (OTU28), *Buttiauxella* (OTU263, 266, 386, 1000, 14,025), *Citrobacter* (OTU66), *Erwinia* (OTU204), *Escherichia/Shigella* (OTU6), *Hafnia* (OTU1, 8, 10), *Kluyvera* (OTU3, 7, 990, 14,028), *Obesumbacterium* (OTU221), *Raoultella* (OTU67), and *Serratia* (OTU5, 22), as well as four unidentified OTUs of Enterobacteriaceae (OTU27, 210, 294, 8791). Within Bacilli, eight OTUs were assigned to the genus *Bacillus* (OTU70), *Enterococcus* (OTU128), *Lactobacillus* (OTU2, 11, 122, 125, 205), and *Pediococcus* (OTU207).

Core OTUs were determined by the shared OTUs in all fecal samples. Twenty-one OTUs (representing 91.13% total reads) were classified into 3 phyla, 4 classes, 5 orders, 8 families, and 13 genera (Table 3).

### 3.4. Alpha and Beta Diversity

Alpha diversity of bacterial communities was assessed in terms of observed number of OTUs, richness estimate (Chao1 and ACE), and diversity index (Shannon and Simpson) (Figure 2, Table 1 and Table 2). In alpha diversity analysis based on the minimal read number (168,005 reads), the total OTU number was 2982 and ranged from 179 to 865 over all samples (Table 4). In general, the bacterial communities were slightly less abundant in early-instar nymph stages as compared to other developmental stages (Table 2). There were no significant differences among eight groups in all four diversity indices according KW rank tests: ACE (Chi-squared = 8.6308, Bonferroni-corrected *p* = 0.2803), Chao1 (Chi-squared = 10.51, Bonferroni-corrected *p* = 0.1615), Shannon (Chi-squared = 6.8006, Bonferroni-corrected *p* = 0.4499), and Simpson (Chi-squared = 7.1247, Bonferroni-corrected *p* = 0.416) (Appendix A). Nonparametric paired *t*-test only revealed significant differences in the Shannon diversity indices between male vs. female (*p* < 2.2 × 10^−16^) and MN12 vs. FN12 (*p* = 0.028) (Appendix A).

Analysis of similarity (ANOSIM) at the OTU level based on Bray-Curtis dissimilarity showed an overall significant difference among developmental stages and sexes (R = 0.148, *p* = 7 × 10^−4^). To better visualize the results, both PCoA and NMDS analyses based on the Bray-Curtis dissimilarity was plotted. The PCoA analyses showed significant differences at the OTU level among groups using two comparisons (ANOSIM: *p* = 7 × 10^−4^ using the Bray-Curtis dissimilarity indices, and *p* = 1 × 10^−4^ using unweighted UniFrac distances); the comparison using weighted UniFrac distances was not significant (*p* = 0.2341) (Figure 3A–C). The NMDS analysis (Stress value = 0.09) failed to provide a clear clustering based on the developmental stages and sexes (Figure 3D). LEfSe analysis identified two families, seven genera, and an unclassified Enterobacteriaceae family to be significantly associated with different development stages and sexes. *Kluyvera* and *Mesorhizobium* were more abundant in the MN12 group. *Megamonas* was more abundant in the MN35 group. The Enterococcaceae family and *Enterococcus* were more abundant in the MA group. The Brucellaceae family, *Serratia*, and *Ochrobactrum* were more abundant in the FN12 group. *Hafnia* was more abundant in the FN67 group. The unclassified Enterobacteriaceae family was more abundant in the FA group (Figure 3E). To demonstrate differences in genus abundance profiles between samples, a heat map was generated with R using the pheatmap package. The relative abundance and distribution of gut bacteria communities at genus level was shown by the heat map (Figure 3F). There were no significant differences in the abundance of fecal bacteria at the genus level among the eight groups. Compared to the *G. gratiosa* nymph group, adults contain more unique OTUs (together constituting 12.2% and 16.5% OTUs in the MA and FA groups, respectively). Beta diversity was further characterized by OTUs shared between groups, revealing that 17.94% OTUs were shared among all eight groups (Figure 1D).

## 4. Discussion

Bacteria can play beneficial, and often essential, roles for host insect survival. The most dominant role of the gut bacteria is essential nutrient provisioning, followed by digestion and detoxification [55]. While insects readily acquire several bacteria from their surroundings during their life cycle, others are vertically transmitted or inherited [56]. In the past decade, the bacterial communities of insects have been extensively studied between wild and laboratory-reared populations [57,58], and gut and fecal samples [59], as well as across different developmental stages [24,29,60,61]. It has been observed that an insect gut microbial community can be influenced by the host’s diet and surroundings. In this study, *G. gratiosa* were raised separately under the same conditions. Therefore, the bacterial communities were mainly affected by the developmental stages and sexes.

Omnivorous insects commonly have a higher gut bacterial diversity [38,62], which could partly be due to their highly diversified diets, as they eat whatever they find. We observed 2982 OTUs in omnivorous katydid *G. gratiosa*, which were assigned into 25 phyla, 42 classes, 60 orders, 116 families, 241 genera, and some unclassified groups. Approximately 30% of OTUs were not assigned to the genus level, suggesting the existence of previously uncharacterized taxa [63]. Similar higher gut bacterial diversity has been reported in the omnivorous cockroach *Blattella germanica* (2713 OTUs) [12]. It is worth noting that omnivorous cockroaches host a diverse gut microbiome that is not essential for host survival [10]. Most gut microbiota of omnivorous cockroaches come from their surroundings rather than vertical transmission [10].

An insect bacterial community is usually dominated by few phyla, such as Proteobacteria, Firmicutes, and Bacteroidetes [32,64,65]. Although the OTUs in *G. gratiosa* feces were assigned into 25 bacterial phyla, the majority of OTUs (91.18%) and reads (99.81%) are concentrated in Proteobacteria and Firmicutes. The phylum Proteobacteria and Firmicutes were also the most common found in other orthopteran species [36,38]. Proteobacteria were the mostly transient microbes of the variegated grasshopper *Zonocerus variegatus* [66]. Although male grasshoppers showed a significantly higher alpha diversity, the prevalence of the main phyla does not shift [36]. The Proteobacteria phylum is highly diverse and contains a large variety of species that are adapted to several environments [67]. Firmicutes and Proteobacteria play important roles in maintaining the growth and development of insects during the metabolism of secondary metabolites in host plants [39].

The families Clostridiales, Desulfovivrionales, and Bacteriodales are common to omnivorous cockroaches [11,68,69]. Enterobacteriaceae often represent one of the dominant bacterial families in insect guts [64,70]. The gut microbiota of *G. gratiosa* was dominated by *Kluyvera*, *Obesumbacterium*, *Buttiauxella*, *Lactobacillus*, and *Hafnia* genera. The gut microbiota of *Apis cerana* is dominated by *Serratia* (Proteobacteria: Enterobacteriaceae), *Snodgrassella* (Proteobacteria: Neisseriaceae), and *Lactobacillus* (Firmicutes: Lactobacillaceae) genera [71].

The gut core microbiota are quite stable over evolutionary time [72] and likely have an important role in host nutrition [73]. In contrast to higher bacterial richness (2982 OTUs), only 21 core OTUs were presented in all *G. gratiosa* fecal samples. Similar results (25 of 2713 bacterial OTUs) have been found in omnivorous cockroach *B. germanica* [12]. Only six core bacteria OTUs were shared across different developmental stages of *Bactrocera minax* (Diptera: Tephritidae) [70]. This suggests the possibility of vertical transmission of these core bacteria [70]. The variability of microbial composition within groups could be the random acquisition of microorganisms from surroundings [74]. Some of these core OTUs have a low abundance, which may become dominant in response to environmental changes and play a key role in the ecosystem [12,75].

The dietary change in different developmental stages played a substantial role in shaping bacterial communities of holometabolic insects [28,59,76,77,78,79]. For example, the bacterial diversity of *Chironomus transvaalensis* declines as it evolves from egg mass to adult, while the highest richness was observed in the pupal stage [60]. The proportion of each intestinal microbe is related to the host development process and gender, indicating that the physiological changes in the host affect the growth of microorganisms. Orthopterans are hemimetabolous insects with similar nutritional environments and requirements during their developmental stages. The bacterial richness of the *G. gratiosa* early-instar nymph is very similar to other developmental stages. The influence of sex is minimal on the insect bacterial composition [80,81]. Male grasshoppers have a significantly higher alpha diversity than the females, which could be attributed to rare taxa rather than main phyla [36]. It is therefore not surprising that minimal variation in *G. gratiosa* bacterial communities was observed among differential developmental stages and sexes.

The symbiotic association was due to the influence of diet, but also may have been related to vertical transmission [82]. Symbiotic bacteria of insects have also been reported to be transmitted horizontally and vertically [83,84]. The majority of symbionts are transmitted from the mother to the offspring vertically. Previous studies have revealed the host insects generally transmit their symbionts vertically by egg surface contamination, coprophagy, or symbiont capsule provisioning, etc. Transmission through eggs requires that larvae take up these bacteria when hatching. The fecal–oral route, early-instar nymph eating mother’s feces, may be key to a gut microbial community establishing in German cockroach *B. germanica* [85]. Newly hatched larvae take up symbionts maternally transferred via biting through and fully ingesting eggshells contaminated with bacteria, e.g., smeared over the egg surface following oviposition [6,17,86]. Some endophytic bacteria in plant-feeding insect gut were the same as plant tissues, suggesting that their microbial communities might be established by acquiring bacteria from plant tissues [17].

We assume that *G. gratiosa* acquired bacteria vertically by feeding on contaminated eggshells. Insect gut communities are dominated by widely distributed bacteria that appear to colonize hosts opportunistically [1]. The taxonomic composition of egg microbiota revealed a relatively high abundance of Proteobacteria, which represents the maternal and environmental sources of gut bacteria [6]. The horizontal transmission of microbial communities has been demonstrated in hemimetabolous *Scaphoideus titanus* [87]. The intestinal environment of *G. gratiosa* has strict limits on bacteria survival. Only a few bacteria can adapt to the intestinal environment, resulting in the diversity of intestinal microbe being much lower than the surroundings. Symbionts were readily acquired horizontally when *Pyrrhocoris apterus* nymphs were reared in the presence of symbionts on the surface of the eggs during oviposition, feces, or adult bugs [16].

## 5. Conclusions

Bacterial communities present in laboratory-reared *G. gratiosa* feces were studied using Illumina sequencing of the 16S rDNA V3-V4 region. *G. gratiosa* is an omnivorous and hemimetabolous insect, which passes through seven nymphal stages from egg to adult. Considering *G. gratiosa* were raised separately under the same conditions, the bacterial communities were not affected by diet and rearing conditions. The early-instar nymphs harbored a similar bacterial community with other developmental stages, which contain higher species diversity. Both PCoA and NMDS analysis at the OTU level failed to provide a clear clustering based on the developmental stages and sexes. One factor could be the extent to which the feeding ecology of hemimetabolous insects is very similar from early-instar nymphs up to adults. This proves that there is vertical transmission of microorganisms, early-instar nymphs eating eggshells to obtain the flora. These data will provide an overall view of the bacterial community in the feces across *G. gratiosa* life stages and sexes.

## Figures and Tables

**Figure 1 insects-13-00361-f001:**
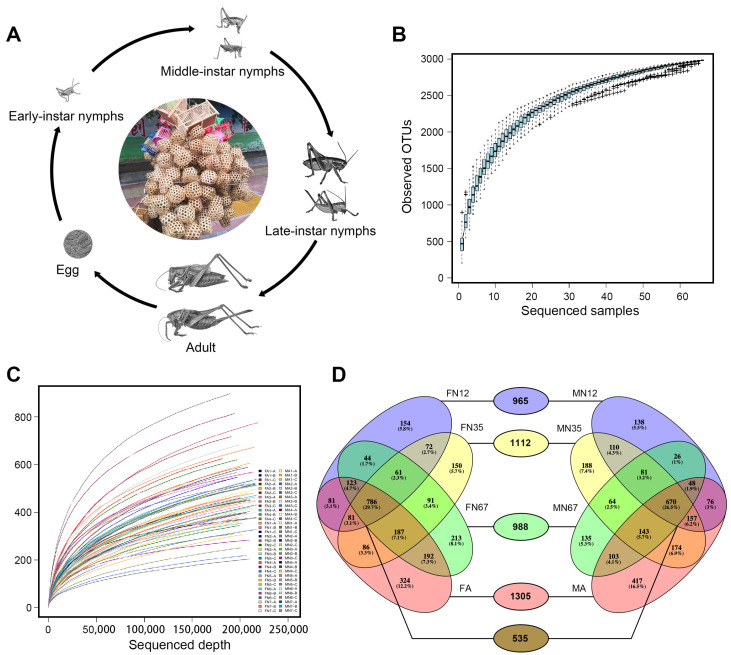
Changes in bacterial community diversity across life stages of *Gampsocleis gratiosa*. (**A**) Overview of development stages of *Gampsocleis gratiosa*, (**B**) species accumulation curves, (**C**) rarefaction analysis for each sample, (**D**) Venn diagram of OTU distribution.

**Figure 2 insects-13-00361-f002:**
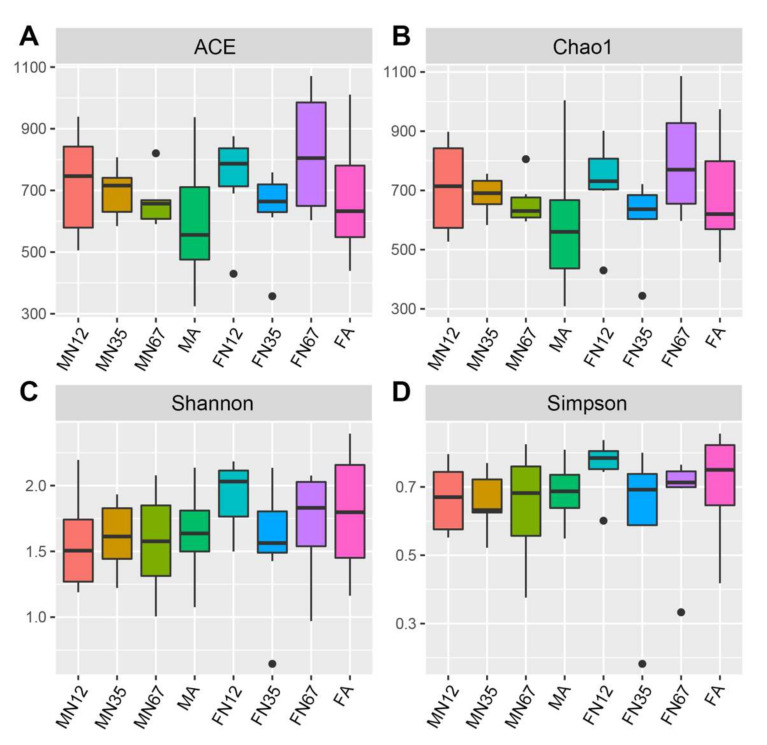
The alpha diversity of bacterial composition in *Gampsocleis gratiosa* fecal samples. (**A**) ACE, (**B**) Chao1, (**C**) Shannon, (**D**) Simpson.

**Figure 3 insects-13-00361-f003:**
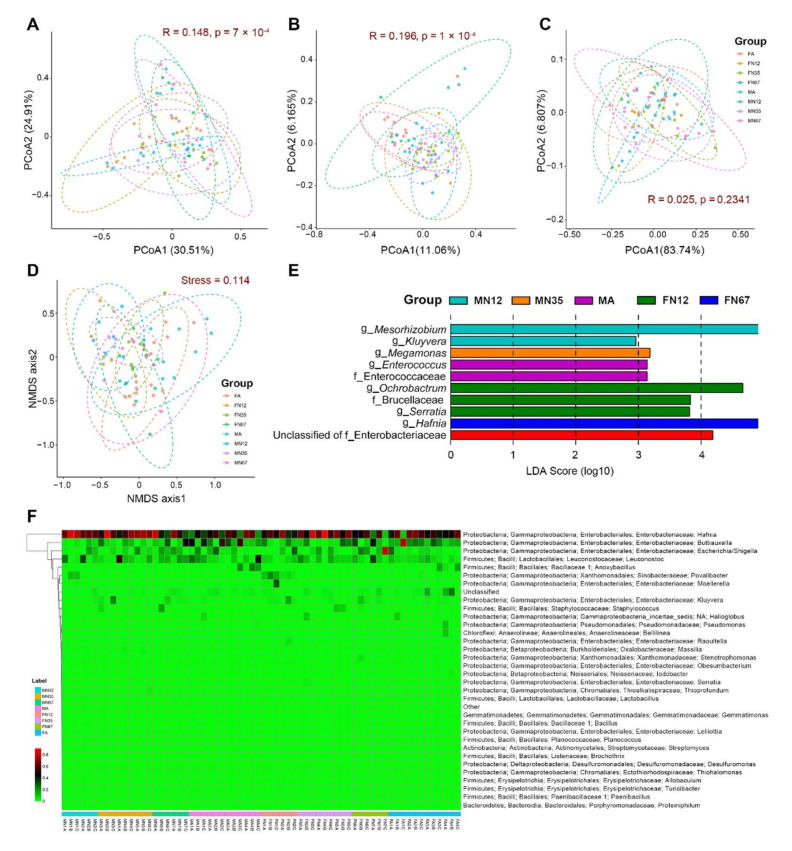
Bacterial beta diversity of *Gampsocleis gratios*. (**A**–**C**) Principal coordinate analysis (PCoA) based on Bray-Curtis dissimilarity indices, unweighted UniFrac, and weighted UniFrac distance. (**D**) Non-metric multidimensional scaling analysis (NMDS) based on Bray-Curtis dissimilarity indices. (**E**) Differences in bacterial taxa among groups determined by linear discriminative analysis effect size (LEfSe), Abbreviation: g_, genus and f_, family. (**F**) Heat map of the relative abundance of bacterial communities at the genus level. The genus of which abundance is less than 0.2% in all samples were classified into “Other”. The OTUs was not classified into genus level were marked by “Unclassified”. Heatmap color scale represents the proportion of sequences assigned to the genus.

**Table 1 insects-13-00361-t001:** The sample labels and group information of this study.

Group	Sex	Developmental Stage	Time Points
MN12	Male	Early nymphs	1st instar male nymphs, MN1
			2nd instar male nymphs, MN2
MN35	Male	Middle nymphs	3rd instar male nymphs, MN3
			4th instar male nymphs, MN4
			5th instar male nymphs, MN5
MN67	Male	Late nymphs	6th instar male nymphs, MN6
			7th instar male nymphs, MN7
MA	Male	Adults	0 day male adult, MA1
			7 day male adult, MA2
			14 day male adult, MA3
			21 day male adult, MA4
FN12	Female	Early nymphs	1st instar female nymphs, FN1
			2nd instar female nymphs, FN2
FN35	Female	Middle nymphs	3rd instar female nymphs, FN3
			4th instar female nymphs, FN4
			5th instar female nymphs, FN5
FN67	Female	Late nymphs	6th instar female nymphs, FN6
			7th instar female nymphs, FN7
FA	Female	Adults	0 day female adult, FA1
			7 day female adult, FA2
			14 day female adult, FA3
			21 day female adult, FA4

**Table 2 insects-13-00361-t002:** No. of OTUs and relative abundance of dominant taxa at different taxonomic levels.

Taxonomic Levels	No. OTUs (%)	% Read Counts
Proteobacteria	2208 (74.04%)	57.22%
Gammaproteobacteria	2083 (69.85%)	57.05%
Enterobacteriales	2000 (67.07%)	56.39%
Enterobacteriaceae	2000 (67.07%)	56.39%
*Kluyvera*	387 (12.98%)	14.62%
*Obesumbacterium*	339 (11.37%)	0.26%
*Buttiauxella*	296 (9.93%)	1.01%
*Hafnia*	152 (5.10%)	33.17%
*Serratia*	75 (2.52%)	1.98%
*Raoultella*	44 (1.48%)	0.16%
Alphaproteobacteria	57 (1.91%)	0.08%
Betaproteobacteria	35 (1.17%)	0.08%
Firmicutes	511 (17.14%)	42.59%
Bacilli	418 (14.02%)	42.52%
Lactobacillales	367 (12.31%)	41.83%
Lactobacillaceae	300 (10.06%)	41.34%
*Lactobacillus*	286 (9.59%)	40.71%
Clostridia	81 (2.72%)	0.06%
Bacteroidetes	58 (1.95%)	0.04%
Bacteroidia	41 (1.37%)	0.03%
Acidobacteria	55 (1.84%)	0.01%

**Table 3 insects-13-00361-t003:** Abundance and classification of core OTUs.

OTU ID	% Read Counts	Phylum	Class	Order	Family	Genus
Otu128	0.14%	Firmicutes	Bacilli	Lactobacillales	Enterococcaceae	Enterococcus
Otu2	38.95%	Firmicutes	Bacilli	Lactobacillales	Lactobacillaceae	Lactobacillus
Otu408509	0.04%	Firmicutes	Bacilli	Lactobacillales	Lactobacillaceae	Lactobacillus
Otu95456	0.02%	Firmicutes	Bacilli	Lactobacillales	Lactobacillaceae	Lactobacillus
Otu211	0.09%	Firmicutes	Bacilli	Lactobacillales	Leuconostocaceae	Weissella
Otu125	0.18%	Firmicutes	Bacilli	Lactobacillales	Streptococcaceae	Lactococcus
Otu107	0.11%	Firmicutes	Bacilli	Lactobacillales	Streptococcaceae	Lactococcus
Otu92	0.04%	Firmicutes	Bacilli	Lactobacillales	Streptococcaceae	Lactococcus
Otu212	0.06%	Proteobacteria	Betaproteobacteria	Burkholderiales	Burkholderiaceae	Burkholderia
Otu14025	0.21%	Proteobacteria	Gammaproteobacteria	Enterobacteriales	Enterobacteriaceae	Buttiauxella
Otu66	1.48%	Proteobacteria	Gammaproteobacteria	Enterobacteriales	Enterobacteriaceae	Citrobacter
Otu6	0.82%	Proteobacteria	Gammaproteobacteria	Enterobacteriales	Enterobacteriaceae	Escherichia/Shigella
Otu1	13.90%	Proteobacteria	Gammaproteobacteria	Enterobacteriales	Enterobacteriaceae	Hafnia
Otu8	10.12%	Proteobacteria	Gammaproteobacteria	Enterobacteriales	Enterobacteriaceae	Hafnia
Otu10	7.70%	Proteobacteria	Gammaproteobacteria	Enterobacteriales	Enterobacteriaceae	Hafnia
Otu3	15.26%	Proteobacteria	Gammaproteobacteria	Enterobacteriales	Enterobacteriaceae	Kluyvera
Otu22	0.65%	Proteobacteria	Gammaproteobacteria	Enterobacteriales	Enterobacteriaceae	Serratia
Otu27	0.87%	Proteobacteria	Gammaproteobacteria	Enterobacteriales	Enterobacteriaceae	Unclassified
Otu210	0.41%	Proteobacteria	Gammaproteobacteria	Enterobacteriales	Enterobacteriaceae	Unclassified
Otu123	0.07%	Proteobacteria	Gammaproteobacteria	Pseudomonadales	Pseudomonadaceae	Pseudomonas
Otu95457	0.02%	Thermotogae	Thermotogae	Petrotogales	Petrotogaceae	Defluviitoga

**Table 4 insects-13-00361-t004:** Alpha diversity indices (Mean ± SD) of bacterial community in feces across *Gampsocleis gratiosa*’s different developmental stages and sexes.

Group	Sample Size	Number of OTUs	Chao1	ACE	Shannon	Simpson
MN12	6	475.667 ± 109.432	710.841 ± 159.337	722.539 ± 174.095	1.568 ± 0.383	0.667 ± 0.103
MN35	9	459.333 ± 40.765	682.463 ± 61.026	705.487 ± 78.775	1.602 ± 0.262	0.659 ± 0.078
MN67	6	396.667 ± 78.194	659.140 ± 78.934	664.879 ± 83.689	1.567 ± 0.403	0.644 ± 0.168
MA	12	355.500 ± 153.390	565.359 ± 193.420	582.558 ± 183.128	1.620 ± 0.318	0.683 ± 0.077
FN12	6	512.000 ± 83.816	720.064 ± 161.132	736.850 ± 163.78	1.927 ± 0.275	0.760 ± 0.084
FN35	9	419.444 ± 61.561	615.366 ± 110.787	647.467 ± 119.32	1.577 ± 0.418	0.632 ± 0.185
FN67	6	592.000 ± 179.861	803.358 ± 194.458	821.251 ± 202.958	1.707 ± 0.430	0.663 ± 0.163
FA	12	446.833 ± 157.623	663.298 ± 163.211	667.523 ± 173.058	1.812 ± 0.408	0.714 ± 0.134

## Data Availability

The datasets generated during this study were deposited in NCBI under Bio-project PRJNA795812 (Bio-sample accessions SAMN24777588–SAMN24777653, SRA accession SRR17518483–SRR17518548).

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
