# Peer review of "Bacterial Communities in the Feces of Laboratory Reared Gampsocleis gratiosa (Orthoptera: Tettigoniidae) across Different Developmental Stages and Sexes"

_insects, 2022, doi:10.3390/insects13040361_

Round 1

Reviewer 1 Report

In this submission, the authors sought to investigate the differences across various developmental stages of the lab-reared Katydid, Gampsocleis gratiosa. The data presented , while interesting could be presented in a more appealing and direct manner and discussed more succintly. Below, I highlight in detail some of my concerns about the submission.

Line 10: Be specific . Harbor microbial community where?

Line 26: Where is this conclusion coming from given the lack of clustering? 

Line 32: Again, be specific about where this microbial community is . Gut, exoskeleton, ovaries, feaces?

Lines 43-45: This is not true. 

Line 45-46:Be specific. Fix this issue throughout the manuscript. Also this sentence doesnt seem to have any relevance to the sentences above it nor the next paragraph.

Overall, the introduction  could be restructured.

Lines 84-89: You would expect or you expected. Rewrite the hypothesis here. Whats the bases for having any expectations regarding the core gut microbiome?

Materials and Methods

Line 133-136: Did you use SILVA or RDP?The confidence threshold you use for either of these databases would be different. As is, it reds like you used both. which cant be the case. Which versions of these databases did you use? Fix this error.

Results

Limes 161: Table 1 can be presented as a supplementary material or if it has to stay, what are we looking at? The summary of the quality filtering steps or the results form the alpha diversity analyses? If it is alpha diversity analyses, which ones are significantly different? Redo the  table .

 Lines 162:

80% read similarity to the RDP database base don 97% clustering of OTUs.

 Why not use  97% clustering both for OTU classification as well as for similarity to database? especially if you also used SILV, which makes it confusing.

Lines 175-199: Since it is not clear if these community diversity summaries are based on  rarefied data, please remove this section or specify what thje rarefaction limit was . In section 3.4, you state that the rarefaction limit was 16805, so thats ok for the diversity analyses reported. However for sections 3.1 -3.3 you dont mention if this was with rarefied data or not so remove or address this.

Lines 242-243: If there were no significant differencs, then everything in lines 232-248 is redundant. Just present the table and show significant differences or not with letters across the experimental groups.

Lines 250-266: Do you have pairwise comparisons of the beta diversity analyses for the 8 experimental groups? EG. 1st instars vs adults

Discussion

This section can be shortened considerably!! No need to  compare lab and field as you did not examine this.. Stick to insect i atleast the same order as the insects you studied. You are going across so many different insect and orders, its hard to track what the point or insinuation is regarding gut microbial composition and/or function. You focused on a hemimetabolous insect, thus lines 334-344 are a bit irrelevant and pointless

Author Response

Thank you very much for your suggestions and giving us an opportunity to revise our manuscript. The point-to-point response is as follows:

  1. Line 10: Be specific. Harbor microbial community where?

Thanks. In revision, we have modified it as “Many insects host a diverse gut microbial community, …”.

  1. Line 26: Where is this conclusion coming from given the lack of clustering?

Thanks. In revision, we have modified it as “Overall, we assume that G. gratiosa transmits bacteria vertically by eating contaminated eggshells, and both developmental stages and sexes had no significant effect on fecal bacterial community.

  1. Line 32: Again, be specific about where this microbial community is. Gut, exoskeleton, ovaries, feaces?

Thanks. In revision, we have modified it as “Many insects harbor a diverse gut microbial community, including protists, fungi, archaea, and bacteria [1]”.

  1. Lines 43-45: This is not true.

Thanks. In revision, we have removed it.

  1. Line 45-46: Be specific. Fix this issue throughout the manuscript. Also this sentence doesn’t seem to have any relevance to the sentences above it nor the next paragraph.

Thanks. In revision, we have removed it.

  1. Overall, the introduction could be restructured.

Thanks. In revision, we have restructured the introduction.

  1. Lines 84-89: You would expect or you expected. Rewrite the hypothesis here. Whats the bases for having any expectations regarding the core gut microbiome?

Thanks. In revision, we have modified it as “We hypothesize that bacterial diversity increased as G. gratiosa progressed through 1st instar nymph to adult, and shared several core bacteria between different life stages and sexes.

Materials and Methods

  1. Line 133-136: Did you use SILVA or RDP? The confidence threshold you use for either of these databases would be different. As is, it reads like you used both. which cant be the case. Which versions of these databases did you use? Fix this error.

Thanks. In revision, we have modified it as “The most abundant sequence of each OTU was chosen as a representative sequence, and taxonomic assignment was made with the Bayesian RDP classifier 2.12 against the Ribosomal Project Database (Confidence level = 0.8) [47] and Silva databases (Similarly > 90% and Coverage > 90%) [48].

Results

  1. Limes 161: Table 1 can be presented as a supplementary material or if it has to stay, what are we looking at? The summary of the quality filtering steps or the results form the alpha diversity analyses? If it is alpha diversity analyses, which ones are significantly different? Redo the table.

Thanks. In revision, we have given it as supplementary material (Table S1).

  1. Lines 162: 80% read similarity to the RDP database base don 97% clustering of OTUs. Why not use 97% clustering both for OTU classification as well as for similarity to database? especially if you also used SILV, which makes it confusing.

Thanks. See above Line 133-136.

  1. Lines 175-199: Since it is not clear if these community diversity summaries are based on rarefied data, please remove this section or specify what the rarefaction limit was. In section 3.4, you state that the rarefaction limit was 16805, so thats ok for the diversity analyses reported. However for sections 3.1 -3.3 you dont mention if this was with rarefied data or not so remove or address this.

Thanks. We used original data in sections 3.1-3.3. Rarefied data only used when comparing the diversity index among the different groups.

  1. Lines 242-243: If there were no significant differencs, then everything in lines 232-248 is redundant. Just present the table and show significant differences or not with letters across the experimental groups.

Thanks. In revision, we have removed “Two richness estimates, Chao1 varied from 565.359±193.42 in MA to 803.358±194.458 in FN67 group, and ACE varied from 582.558±183.128 in MA to 821.251±202.958 in FN67 group. Two diversity indices, Shannon varied from 1.567±0.403 in MN67 to 1.927±0.275 in FN12 group, and Simpson varied from 0.632±0.185 in FN35 to 0.760±0.084 in FN12 group.”

  1. Lines 250-266: Do you have pairwise comparisons of the beta diversity analyses for the 8 experimental groups? EG. 1st instars vs adults

Thanks. We did not do pairwise comparisons of the beta diversity analyses.

Discussion

  1. This section can be shortened considerably!! No need to compare lab and field as you did not examine this. Stick to insect at least the same order as the insects you studied. You are going across so many different insect and orders, its hard to track what the point or insinuation is regarding gut microbial composition and/or function. You focused on a hemimetabolous insect, thus lines 334-344 are a bit irrelevant and pointless.

Thanks. In revision, we have restructured the discussion.

Reviewer 2 Report

This study clarified the bacterial community in feces among developmental stages and sexes of the laboratory-maintained katydids, Gampsocleis gratiosa. Among 2982 OTUs found these analyses, Bray-Curtis did not show clear clustering and only 21 OTUs were shared among examined samples. Nevertheless, the bacterial community was not greatly changed across the developmental stages and sexes. This paper does not greatly promote our understanding on insect physiology and ecology, but it may be more relevant to microbiology.

General remarks:

Because the results were obtained from the laboratory-maintained individuals that have limited contacts to microorganisms, it was unclear whether the results were artificial and how much the obtained knowledge can be applicable to wild insects that may contact more diverse microorganisms. I strongly encourage the authors to collect some wild insects of different developmental stages and compared their bacterial community with that of the laboratory-maintained individuals. Otherwise, it is uncertain whether the wild individuals also harbor the similar bacterial community to the laboratory-maintained ones and whether bacterial members consisting of the core OTUs in the laboratory-maintained insects are also present in the wild individuals.

It was also unclear whether the katydids depend on the gut microbiota. Do the katydids harbor a significant number of bacterial cells in the gut? Was there any study suggesting the role of the gut bacteria in the growth of this insect? Without these introductions, it was unclear how much this study was worth to do.

The presentation in this manuscript were very complicated and hard to follow because detailed information was only described in the main text. Moreover, the authors ceaselessly mentioned the abundance of each bacterial taxon or OTU, but its importance may not be understood by entomologists who are not familiar with bacteria. Was there any significant difference from other insect gut microbiota?

Specific comments:

L33: Some insects such as xylophagous insects harbor numerous bacterial cells in the guts, but I don’t think many insects harbor microorganisms that outnumber their own cells.

L36 and elsewhere: You sometimes mention only scientific names to indicate certain species of insects, but most readers may not be familiar with scientific names if they are out of expertise. Please indicate common names for them (e.g. weevils for Curculio chinensis).

L41: Cryptocercus punctulatus is not omnivorous, but it is a xylophagous cockroach.

L43-L45: What is a definition of a resident gut microbiome? Although caterpillars do not harbor a gut microbiome, in most cases, insects may acquire bacteria from the environment, which settle in the gut. Why were stinkbugs and wood-feeding beetles excluded?

L60: “Bacteria was the predominant microbiota in all insect gut”. This is incompatible with the above sentence (L43-45). Moreover, compared to mammals such as humans, the majority of insects harbor lesser number of bacteria in the gut. Diversity of bacteria in the gut of insects is often very low.

L96: Replace Eppendorf tube with a microtube.

L97: Rephrase “Try to”.

L100-L109: The sample labels are very complicated, and the relationship between each sample and each group is hard to follow. Please make a table to clarify this relationship.

L114: “Takahashi et al., 2014” should be numbered and listed in the references. In addition, please indicate the adaptor sequences and how the adaptors were ligated. If you use a kit for the library preparation, please specify.

L123-L125: Did you merge the paired-end reads without QC? Usually, quality of reverse reads generated from MiSeq are fairly low. What was the method or program for quality filtering?

L132: Describe the method by which you removed organelle and nonbacterial sequence.

L147-L148: Why did you analyse only Bray-Curtis similarity? I encourage you to plot PCoA on weighted and unweighted UniFrac.

L181: Relative abundance of bacterial taxa in each sample should be provided as a supplementary table. (I’m assuming a table similar to Table S1 in Dietrich et al. (2014) Appl. Environ. Microbiol. 80, 2261-2269).

L184-L230: These descriptions with percentage of OTUs or out of total reads were very hard to follow. Please arrange them into tables. Especially, the core OTUs and their abundance need to be shown in a table.

Figure 3C: “LDA Score (log10)” should be placed more closely to the bar graph.

Figure 3D: The vertical axis only shows names of genera. Please indicate all relevant taxonomical information such as phyla, classes, orders, and families as well. Please indicate the unit for the scale. What do you want to mention based on this heatmap?

L358-L360: B. germanica is not a social insect.

L366: Did you perform any observation that suggested eggshell-eating behaviors by this katydid. Otherwise, it is just a speculation to mention it rather than assumption. It is also possible that preferred microorganisms are deterministically acquired from the environment as previously demonstrated using Shelfordella lateralis (Mikaelyan et al. 2016, Appl. Environ. Microbiol. 82, 1256-1263).

Ref. 18: Clarify the page/publication number.

Author Response

Thank you very much for your suggestions and giving us an opportunity to revise our manuscript. The point-to-point response is as follows:

General remarks:

1. Because the results were obtained from the laboratory-maintained individuals that have limited contacts to microorganisms, it was unclear whether the results were artificial and how much the obtained knowledge can be applicable to wild insects that may contact more diverse microorganisms. I strongly encourage the authors to collect some wild insects of different developmental stages and compared their bacterial community with that of the laboratory-maintained individuals. Otherwise, it is uncertain whether the wild individuals also harbor the similar bacterial community to the laboratory-maintained ones and whether bacterial members consisting of the core OTUs in the laboratory-maintained insects are also present in the wild individuals.

Thanks. The population size of this insect is very small. It's hard to collect wild individuals of different developmental stages.

2. It was also unclear whether the katydids depend on the gut microbiota. Do the katydids harbor a significant number of bacterial cells in the gut? Was there any study suggesting the role of the gut bacteria in the growth of this insect? Without these introductions, it was unclear how much this study was worth to do.

Thanks. Insect guts present distinctive environments for microbial colonization. These microbiotas played an important role in nutrient uptake and adaptability to the environment, and were directly related to the growth, development, and reproduction of the insects. Omnivorous insects contain relatively rich gut microbial species. Gampsocleis gratiosa (Orthoptera, Tettigoniidae) is an economically important singing pet in China.

3. The presentation in this manuscript were very complicated and hard to follow because detailed information was only described in the main text. Moreover, the authors ceaselessly mentioned the abundance of each bacterial taxon or OTU, but its importance may not be understood by entomologists who are not familiar with bacteria. Was there any significant difference from other insect gut microbiota?

Thanks. The early-instar nymphs of Gampsocleis gratiosa harbored similar bacterial community with other developmental stages. Relative to large abundance of gut microbes (2982 OTUs), only 21 core OTUs were shared by all fecal samples. Similar results have been found in other omnivorous insects.

4. Specific comments:

1)L33: Some insects such as xylophagous insects harbor numerous bacterial cells in the guts, but I don’t think many insects harbor microorganisms that outnumber their own cells.

Thanks. In revision, we have deleted it.

2)L36 and elsewhere: You sometimes mention only scientific names to indicate certain species of insects, but most readers may not be familiar with scientific names if they are out of expertise. Please indicate common names for them (e.g. weevils for Curculio chinensis).

Thanks. In revision, we have given common names.

3)L41: Cryptocercus punctulatus is not omnivorous, but it is a xylophagous cockroach.

Thanks. In revision, Cryptocercus punctulatus has been deleted.

4)L43-L45: What is a definition of a resident gut microbiome? Although caterpillars do not harbor a gut microbiome, in most cases, insects may acquire bacteria from the environment, which settle in the gut. Why were stinkbugs and wood-feeding beetles excluded?

Thanks. In revision, the introduction has been restructured based on the suggestions of another Reviewer.

5)L60: “Bacteria was the predominant microbiota in all insect gut”. This is incompatible with the above sentence (L43-45). Moreover, compared to mammals such as humans, the majority of insects harbor lesser number of bacteria in the gut. Diversity of bacteria in the gut of insects is often very low.

Thanks. In revision, we have modified it as “Bacterial species comprise all or most organisms of most insect guts [1].”

6)L96: Replace Eppendorf tube with a microtube.

Thanks. We have replaced Eppendorf tube with a microtube.

7)L97: Rephrase “Try to”.

Thanks. In revision, we have rewritten it as “Avoid interrupting G. gratiosa molting when collecting.”

8)L100-L109: The sample labels are very complicated, and the relationship between each sample and each group is hard to follow. Please make a table to clarify this relationship.

Thanks. In revision, the sample labels and grouping were shown in Table 1.

9)L114: “Takahashi et al., 2014” should be numbered and listed in the references. In addition, please indicate the adaptor sequences and how the adaptors were ligated. If you use a kit for the library preparation, please specify.

Thanks. In revision, we have modified it as numbered and listed in the references. Both barcode and Illumina adaptor sequences are included in primers. In revision, we have rewritten it as “Both primers contained Illumina adapters, and the reverse primer contained a 6-bp barcode sequence unique to each sample.”

10)L123-L125: Did you merge the paired-end reads without QC? Usually, quality of reverse reads generated from MiSeq are fairly low. What was the method or program for quality filtering?

Thanks. The paired-end reads were filtered using Prinseq 0.20.4. In revision, we have rewritten it as “The sequences were filtered by quality using Prinseq 0.20.4 [42], a Phred quality cutoff value of 20 (Q20) and a minimum cut length of 200 nucleotides were used for both strands.”

11)L132: Describe the method by which you removed organelle and nonbacterial sequence.

Thanks. Organelle and nonbacterial sequences were removed based on the results of OTUs taxonomic assignment. In revision, we have rewritten it as “We removed the sequences from OTU tables, which were annotated as chloroplasts or mitochondria, and not assigned to any kingdom, to ensure only bacterial 16S rDNA sequences were included in downstream analyses.”

12)L147-L148: Why did you analyse only Bray-Curtis similarity? I encourage you to plot PCoA on weighted and unweighted UniFrac.

Thanks. In the revision, we have added PCoA analyses based on weighted and unweighted UniFrac distance.

13)L181: Relative abundance of bacterial taxa in each sample should be provided as a supplementary table. (I’m assuming a table similar to Table S1 in Dietrich et al. (2014) Appl. Environ. Microbiol. 80, 2261-2269).

Thanks. In the revision, relative abundance of bacterial taxa in each sample was given as a supplementary Table S2.

14)L184-L230: These descriptions with percentage of OTUs or out of total reads were very hard to follow. Please arrange them into tables. Especially, the core OTUs and their abundance need to be shown in a table.

Thanks. In the revision, we have added Table 3. Abundance and classification of core OTUs.

15)Figure 3C: “LDA Score (log10)” should be placed more closely to the bar graph.

Thanks. In the revision, we have modified.

16)Figure 3D: The vertical axis only shows names of genera. Please indicate all relevant taxonomical information such as phyla, classes, orders, and families as well. Please indicate the unit for the scale. What do you want to mention based on this heatmap?

Thanks. In the revision, we have added all relevant taxonomical information and the unit for the scale. There were no significant differences in the abundance of fecal bacteria at the genus level among the eight groups.

17)L358-L360: B. germanica is not a social insect.

Thanks. In the revision, we have modified.

18)L366: Did you perform any observation that suggested eggshell-eating behaviors by this katydid. Otherwise, it is just a speculation to mention it rather than assumption. It is also possible that preferred microorganisms are deterministically acquired from the environment as previously demonstrated using Shelfordella lateralis (Mikaelyan et al. 2016, Appl. Environ. Microbiol. 82, 1256-1263).

Thanks. Yes, we have observed Gampsocleis gratiosa nymph eggshell-eating behaviors.

19)Ref. 18: Clarify the page/publication number.

Thanks. In the revision, we have carefully checked all references.

Round 2

Reviewer 1 Report

Authors have responded to my initial comments and notes

Author Response

Thanks. I am truly grateful for all your comments and notes, which is helpful for improving our manuscript.

Reviewer 2 Report

The manuscript has been revised adequately, but it is still not clear whether the katydids depend on the gut microbiota. Some insects are independent of the gut bacteria, while no experimental evidence showing dependence of katydids on the gut microbiota has been provided. Thus, a link between the repertoires of the gut microbiota and the function is totally obscure.

The results derived from the lab-maintained katydids might have been artificial, and it is also unclear how much the obtained knowledge can be applicable to wild insects that may contact more diverse microorganisms. 

Table 3 contains many inappropriate line breaks in the middle of taxonomy names. Please correct them properly.

Author Response

1. The manuscript has been revised adequately, but it is still not clear whether the katydids depend on the gut microbiota. Some insects are independent of the gut bacteria, while no experimental evidence showing dependence of katydids on the gut microbiota has been provided. Thus, a link between the repertoires of the gut microbiota and the function is totally obscure.

Thanks. I am truly grateful for all your comments and notes, which is helpful for improving our manuscript. The insect gut microbiota has been shown to enhance a certain enzyme activity or expand the digestion range. Previous studies have shown that many omnivorous insects harbor a diverse gut microbial community. G. gratiosa was omnivorous. Thus, we speculate that G. gratiosa depend on the gut microbiota.

2. The results derived from the lab-maintained katydids might have been artificial, and it is also unclear how much the obtained knowledge can be applicable to wild insects that may contact more diverse microorganisms.

Thanks. We cannot eliminate this situation completely. Previous studies have shown that microbial community shifts between the wild and laboratory-reared ghost moth Thitarodes larvae. As a non-model insect, this study on G. gratiosa is only the beginning. Considering G. gratiosa were raised separately under the same conditions, the bacterial communities were not affected by diet and rearing condition.

3. Table 3 contains many inappropriate line breaks in the middle of taxonomy names. Please correct them properly.

Thanks. In the revision, we have modified. In fact, these inappropriate line breaks do not exist in the Table 3. These inappropriate line breaks are due to the automatic adjustment of Word.